# Organic–Inorganic Hybrid Pigments Based on Bentonite: Strategies to Stabilize the Quinoidal Base Form of Anthocyanin

**DOI:** 10.3390/ijms24032417

**Published:** 2023-01-26

**Authors:** Robson V. Cunha, Alan I. S. Morais, Pollyana Trigueiro, João Sammy N. de Souza, Dihêgo H. L. Damacena, Luciano C. Brandão-Lima, Roosevelt D. S. Bezerra, Maria Gardennia Fonseca, Edson C. Silva-Filho, Josy A. Osajima

**Affiliations:** 1Federal Institute of Piauí, Floriano Campus, IFPI, Floriano 64808-475, PI, Brazil; 2LIMAV-Interdisciplinary Advanced Materials Laboratory, PPGCM-Materials Science and Engineering Graduate Program, UFPI-Federal University of Piaui, Teresina 64049-550, PI, Brazil; 3Materials Science and Engineering Postgraduate Program-PPGCM/CCSST, UFMA, Imperatriz 65900-410, MA, Brazil; 4Federal Institute of Education, Science and Technology of Piauí, Teresina-Central Campus, IFPI, Teresina 64000-040, PI, Brazil; 5Research and Extension Center-Fuel and Materials Laboratory (NPE–LACOM), Federal University of Paraíba, João Pessoa 58051-970, PB, Brazil

**Keywords:** anthocyanin, bentonite, organic–inorganic hybrids, photostability

## Abstract

Anthocyanins are one of the natural pigments that humanity has employed the most and can substitute synthetic food dyes, which are considered toxic. They are responsible for most purple, blue, and red pigment nuances in tubers, fruits, and flowers. However, they have some limitations in light, pH, oxygen, and temperature conditions. Combining biomolecules and inorganic materials such as clay minerals can help to reverse these limitations. The present work aims to produce materials obtained using cetyltrimethylammonium bromide in bentonite clay for incorporation and photostabilization of anthocyanin dye. Characterizations showed that the organic molecules were intercalated between the clay mineral layers, and the dye was successfully incorporated at a different pH. Visible light-driven photostability tests were performed with 200 h of irradiation, confirming that the organic–inorganic matrices were efficient enough to stabilize the quinoidal base form of anthocyanin. The pigment prepared at pH 10 was three-fold more stable than pH 4, showing that the increase in the synthesis pH promotes more stable colors, probably due to the stronger intermolecular interaction obtained under these conditions. Therefore, organobentonite hybrids allow to stabilize the fragile color coming from the quinoidal base form of anthocyanin dyes.

## 1. Introduction

Anthocyanins (ACN) are organic substances responsible for most purple, blue, and red colors of flowers and fruits. ACN have antioxidant properties and can produce free radicals. Despite having great potential for use in several areas, the applicability of ACN is limited by its reactivity and degradability [1,2,3]. The main factors influencing the color and stability of ACNs are the pH, temperature, light, oxygen, copigmentation, solvent effect, and structure [2,3].

Once extracted from their natural sources, the possibility of exploitable color nuances is reduced by the instability of the molecular structure of ACNs. When in solution, the flavylium cation, which is the most stable form of ACN, is converted to a quinoid base under neutral to mildly alkaline conditions and finally to the uncolored chalcone via carbinol pseudobase form with pH increase. All these conversions are accompanied by a marked color change, making it very difficult to apply the color range present in the pH medium from near neutral to slightly alkaline, due to the instability of the flavylium cation derivatives formed [1,4].

Several studies have been conducted to produce materials that can be used as a strategy to improve the color stability of ACN. These strategies include inorganic matrices such as clays, mesoporous materials, and zeolites [4,5]. In these materials, organic molecules can be intercalated in confined spaces or adsorbed on the surface, promoting increased stability of these organic molecules [5]. In addition, recent studies have shown that clay minerals, such as montmorillonite, saponite, and palygorskite improve the photo and chemical properties of ACN through the formation of hybrid materials [1,4,6,7].

Among clays, bentonite (Bent) is the raw clay containing at least 50% of smectite, particularly montmorillonite (Mt). In addition, Bent often contains other clay minerals including illite and/or other minerals such as quartz and feldspar [8,9]. Mt is 2:1 phyllosilicate, where the octahedral sheets lie between two tetrahedral sheets. Octahedral sheets have aluminum or magnesium bonded to oxygen or hydroxyl groups, while tetrahedral sheets are formed by silicon bonded to oxygen atoms [10]. Isomorphic substitutions in the octahedral sheets of the Mt result in the presence of Na^+^ or Ca^2+^ interlayer cations to counterbalance the excess negative charge in the lamella. Mt has a high cation exchange capacity and a large surface area, and it can be functionalized with different organic molecules by chemical modification. Thus, due to the structural and physicochemical properties, Mt is used in diverse industrial applications [9,11].

Additionally, Mt can be modified in many ways with organic molecules, resulting in inorganic–organic hybrid materials. Among them, intercalation of surfactants between the layers of Mt increases the interlayer space and is an effective route to prepare new functional materials [9,12]. Intercalated surfactants of Mt are produced through an ion exchange reaction between organic alkyl ammonium salts and the interlayer cations. Cetyltrimethylammonium bromide (CTAB), [(C_16_H_33_)N(CH_3_)_3_]Br is the most commonly used cationic surfactant for Mt intercalation. CTAB is often used to provide the adequate opening of Mt layers and improve the ability of Mt to incorporate new organic molecules in subsequent reactions [8].

Concerning the hybrid formation with ACN and organoclay, LIMA et al. 2020 [1] investigated the stabilization of the ACN quinoidal base form due to interactions with CTA^+^ molecules previously intercalated in the clay mineral saponite. The color of the hybrid pigment they obtained refers to the quinoidal base form even though they used a buffered medium at pH 3. Thus, this work has a new route to stabilize the quinoidal base form using bentonite as mineral clay, in addition to evaluating the synthesis at pH 7 and 10, which are more favorable to the presence of the quinoidal base form; this has not yet been evaluated in previous studies.

The present work aims to produce materials derived from bentonite clay intercalated with surfactant and apply them to incorporate anthocyanin dye at different pH conditions. Finally, it investigates the photostability under visible light of the ACN incorporated into CTAB-expanded clay materials by colorimetric studies.

## 2. Results and Discussion

CHN elemental analysis of the Bent sample resulted in 1.04 ± 0.02, 1.34 ± 0.00, and 1.05 ± 0.03%, respectively. The nitrogen content and exchange ammonium amount was 0.75 ± 0.02, and the calculated cation exchange capacity (CEC) was 75.23 ± 1.789 cmol(+)/kg (Appendix A). The CEC value was close to the bentonite clays obtained in previous studies [12,13].

X-ray diffractometry (XRD) patterns of the raw and modified bentonites are presented in Figure 1. The XRD of the ACN (Figure 1a) is typical of an amorphous material [14]. XRD patterns of the Bent clay in Figure 1b indicated that Mt is the majority phase in the clay sample. Reflections at 2θ = 7.03°, 19.88°, 20.88°, 28.62°, 34.94°, and 61.92° were indexed to Mt. Other reflections at 2θ = 11.74° and 26.74° were associated with muscovite (Mu) and quartz (Qt) phases, respectively. The reflections at 2θ = 7.06° and 61.92° are related to the crystallographic planes (001) and (060), respectively. These planes are characteristic of dioctahedral montmorillonite. Reflection at 2θ = 7.03° was associated to a basal spacing (d_001_) of 1.26 nm [12,13,15,16]. For CTAB/Bent, Figure 1c shows the disappearance of the reflection at 2θ = 11.74° and 28.62° and the displacement of the peak at 2θ = 7.03° to 4.82°, which corresponded to a d value of 1.83 nm. The displacement of the 001 reflections to a lower 2θ value indicated an increase in the basal space of the Mt after incorporating CTAB and confirms that CTAB was successfully intercalated into the Mt interlayer space [9,12,17]. XRD patterns of the materials after the dye incorporation are presented in Figure 1d–f. The initial reflection (001) at 4.82° for Bent was shifted to 4.56°, 4.62°, and 4.48° for CTAB/ACN/Bentx (x is referred to as pH 4, 7, or 10, respectively), representing basal spacing (d_001_) changes from 1.82 nm to 1.93 nm, 1.91 nm, 1.97 nm, respectively. The increases in d_001_ space can indicate the successful intercalation of the ACN in modified Mt at the adopted pH [1,18].

FTIR spectra of the raw and modified clay mineral samples are presented in Figure 2. The FTIR spectrum of the Bent (Figure 2b) showed characteristic Mt bands located at 3647 and 3439 cm^−1^, assigned to OH stretching of the structural hydroxyl groups (AlOH, MgOH) and SiOH or OH of the interlayer water of the Mt, respectively. The band at 1639 cm^−1^ was related to the angular deformation of the interlayer water. The band at 1035 cm^−1^ was attributed to the Si-O stretching of the tetrahedral sheet of the 2:1 phyllosilicate structure. The band at approximately 916 cm^−1^ was related to the Al-O angular deformation. Finally, the bands at 792 and 637 cm^−1^ were related to the presence of quartz and the M-O-Si angular vibration of cations located at octahedral sites [15,19].

In the FTIR spectrum of the CTAB/Bent in Figure 2c, bands of the inorganic structure were maintained, but three new bands appeared at 2917, 2850, and 1479 cm^−1^. The bands at 2917 and 2850 cm^−1^ were assigned to the asymmetrical and symmetrical CH stretching of methylene groups, indicating the presence of alkyl moieties of the CTAB chain. The band at 1479 cm^−1^ was attributed to the N-H deformations in the ammonium group [1,9,20]. FTIR results corroborated the XRD patterns and confirmed that CTAB was incorporated in the Mt phase.

Figure 2a shows the FTIR spectrum of pure ACN. A broad band centered at 3395 and a low-intensity band at 2930 cm^−1^ were assigned to the OH stretching in the phenol group and CH asymmetric stretching, respectively. The band at 1616 cm^−1^ was attributed to the C=C stretching of the aromatic rings, and the band at 1020 cm^−1^ was related to the CCH deformation of the aromatic rings [21].

Figure 2d–f present the FTIR spectra of the materials after the incorporation of ACN at the different pH used in this study. The incorporation of ACN in the CTAB/Bent sample did not result in changes in the spectra of these materials with the presence of the dye, possibly due to the strong absorption associated to CTAB in the same region as the ACN. However, the FTIR spectra of dyed samples was observed in 1633 cm^−1^, which may be related to the hydrogen interactions between the nitrogen and OH groups of the CTAB/Bent and ACN molecules [1,22].

Possible alterations in the morphology of samples after the reactions were monitored by scanning electron microscopy (SEM) analysis. SEM images are in Figure 3. The morphology of raw Bent (Figure 3a) showed a heterogeneous and irregular structure and the presence of some flakes or agglomerates [20,23,24,25]. After intercalation with CTAB, the CTAB/Bent (Figure 3b) continued to have an irregular and heterogeneous morphology. In addition, the morphology of CTAB/Bent displayed a number of flakes or agglomerates of varying sizes as well as cracks and caves. This may have occurred because incorporating CTAB promoted a displacement in the layers of the Mt, which resulted in a disordered agglomeration of platelets in some parts of the sample [20,23,24]. SEM images of the materials after the ACN incorporation are in Figure 3c–e and indicate that morphology in the form of flakes or agglomerates was maintained [25].

TG/DTG curves of the materials are shown in Figure 4. In addition, Table 1 presents the main thermal degradation events of ACN, Bent, and its derivatives. The ACN had two characteristic mass loss events, the first one with a maximum temperature of 78 °C, referring to the elimination of surface water, and the second event with a maximum temperature of 297 °C, referring to the thermal decomposition of ACN [26]. The raw Bent presented three mass loss events that are characteristic of the bentonite clay. The incorporation of CTAB altered the thermal characteristics of the Bent clay, as shown in the TG and DTG of the CTAB/Bent material. The graph illustrates that the degradation event related to the loss of adsorbed water (first event) in the CTAB/Bent material occurred at a lower temperature due to the increased hydrophobicity of the clay by incorporating CTAB. Thus, the CTAB/Bent material has less water content, and the amount of water eliminated reduces the dehydration temperature [27,28].

Furthermore, the CTAB/Bent material presented three more mass loss events after incorporating CTAB. The maximum temperatures of these three degradation events were 260 °C (second event), 326 °C (third event), and 428 °C (fourth event), which are related to the presence of CTAB in the clay structure [29,30]. The dehydroxylation thermal event (fifth event) shifted from 650 °C to 575 °C, with a decrease in intensity, as shown in the DTG. The TGs and DTGs of the CTAB/ACN/Bent-4, CTAB/ACN/Bent-7, and CTAB/ACN/Bent-10 materials showed the same four thermal degradation events as their precursor CTAB/Bent, with slight variations in degradation temperatures and percentages of mass loss due to intermolecular interactions obtained from the incorporation of ACN. Therefore, these events are related to the degradation of both CTAB and ACN. In addition to these four events, all materials had a fifth thermal degradation event in the temperature ranges of 470–645 °C (CTAB/ACN/Bent-4), 468–663 °C (CTAB/ACN/Bent-7), and 462–662 °C (CTAB/ACN/Bent-10). This thermal degradation event is related to the thermal dehydroxylation of the clay that was shifted to lower temperatures after the formation of the hybrid pigment [1,22].

The flavylium cation form of ACN majorly influenced the UV-Vis spectra of ACN solutions at natural and pH 4. When the pH was increased to pH 7 and 10 before hybrid formation, the shift and broadening of the absorption band compared with natural pH indicated the deprotonation of the flavylium cations into the quinoidal base form of ACN. Furthermore, after incorporating ACN in the CTAB/Bent, the intensity of absorption bands decreased, providing the values of adsorption capacity equal to 55.4% (Figure 5c), 26.7% (Figure 5b), and 44.7% (Figure 5a) for the pigments obtained at pH 10, 7, and 4, respectively, suggesting that the ACN molecule was about 20% more adsorbed at pH 10 (Figure 5c) [1,4].

The photostability of the hybrid dyed materials was evaluated under visible light for 200 h by the L*a*b* colors space parameters of solid hybrid pigments measured before and after light exposure, as seen in Figure 5e and Table 2.

The L* (lightness) coordinate can assume values from 0 (black) to 100 (white). The b* (yellowness) coordinate indicates the yellow–blue component of a color, where positive and negative values indicate yellow and blue, respectively. The a* (redness) coordinate indicates the red–green component of a color, where positive and negative values indicate red and green, respectively [31].

The new colors obtained after ACN incorporation at the respective pH can be seen in digital photographs of Figure 5e and as a function of L*a*b* coordinates of CIE color space in Table 2. The different colors with increased pH are a function of the more pronounced influence of the quinoidal base form of ACN. Comparing the hybrid pigment with the CTAB/Bent precursor, the total color differences ΔE* values (Table 2) calculated before the irradiation were higher when the pH increased, probably due to the increased ACN incorporation and quinoidal base form contribution [1,4].

After irradiation, the color of the hybrid pigment changes its initial CIE L*a*b* coordinates, indicating a fade of the respective color (Figure 5e and Table 2). The changes of L*a*b* coordinates were less pronounced for the CTA/ACN/Bent-10, indicating that the pigment formed at the highest pH had the least fading. The ΔE* values can quantitatively monitor these changes after irradiation, which was about 3-fold smaller for the pigment obtained at pH 10 compared to pH 4 (Table 2) and about 4.5-fold lower than the values published by LIMA et al. 2020 (ΔE* = 19.3 for a pigment based on saponite obtained at pH 3) [1], who synthesized their hybrid pigments at a lower pH. This result indicates that the pigment at pH 10 was the most photostable, probably due to the strong intermolecular interactions formed in the pigment obtained at this pH [1,4].

## 3. Materials and Methods

### 3.1. Material and Chemicals

Raw Bent sample, donated by the Bentonisa company (Boa Vista, Paraíba, Brazil), was used as precursor material. Hexadecyltrimethylammonium bromide (CTAB) (Êxodo Científica, Sumaré, Brazil), ammonium acetate (Êxodo Científica, São Paulo, Brazil), anthocyanin (ACN) (elderberry extract—SCIYU BIOTECH CO., LIMITED, Xi’an, China), hydrochloric acid (Dinâmica, São Paulo, Brazil), sodium hydroxide (Êxodo Científica, São Paulo, Brazil), and ethanol (Dinâmica, São Paulo, Brazil) were analytical grade and used without prior purification. Distilled water was used in all preparations

### 3.2. Cation Exchange Capacity (CEC)

The CEC of Bent sample was determined by ion exchange method using ammonium acetate solution [13,32]. The procedure was carried out with following the steps described in Table 3.

### 3.3. Synthesis of the CTAB/Bent Hybrid

The CTAB/Bent hybrid synthesis was performed using CTAB in the proportion of 200% of the CEC of the clay mineral. In the results obtained from the CEC in Section 3.2 (Appendix A), the amount of CTAB mass corresponded to 200% of the CEC of the Bent (calculation available in Appendix A) [12]. According to Brito et al. 2018, this proportion favors better conditioning of CTAB in clay. Reactions were performed following the method described previously [9,12,13]. Thus, 4.0 g of Bent were suspended in 100 mL of CTAB (2.19 g) surfactant and stirred for 24 h at room temperature (25 °C). The resulting solid was centrifuged (5000 rpm for 10 min) and washed with distilled water. Finally, the produced material was dried at 100 °C for 24 h.

### 3.4. Sequential Infusion of ACN to Produce ACN in the CTAB/Bent Hybrid

According to previous studies, the reaction between the dye and CTAB/Bent hybrid was performed at acidic, neutral, and basic pH [1,12,13]. In a typical procedure, 1 g of CTAB/Bent was dispersed in 100 mL of 600 mg L^−1^aqueous dye solution at pH 4.0, 7.0, and 10.0, and the systems were stirred for 4 h at 25 °C. Next, the pH was adjusted with 1.0 mol L^−1^ HCl and/or 1.0 mol L^−1^ NaOH aqueous solutions. The initial pH of the dye solution (without solid) was ∼4.3. Subsequently, the samples were centrifuged at 5000 rpm for 10 min. The solid fraction was washed with distilled water and centrifuged under the same conditions. Finally, the samples were dried in an oven at 50 °C for 24 h. Samples were denominated as CTAB/ACN/Bentx (*x* is referred to as pH 4, 7 or 10, respectively) (Table 4). The procedure is summarized in the scheme of Figure 6 [33,34].

### 3.5. Photostability Test

The photostability of dye in the CTAB/ACN/Bentx hybrids was performed by exposing the samples to visible light irradiation by using a 160W Hg lamp. The radiation was monitored by a radiometer (Hanna), and the value was 1.55 klux. Each hybrid pigment powder was dispersed in Petri dishes, placed at a distance of 15 cm below the lamp, and irradiated for 200 h [4]. The “Commission Internationale of l’Eclairage” (CIE) 1976 color space system was applied to evaluate the color of the pigments. Measurements were performed as a function of L*, a*, and b* coordinates obtained from a UV-Vis spectrometer, Shimadzu UV-2600i. The differences in colors between unexposed and exposed samples were calculated by $\Delta E*=\sqrt{\left({\left(\Delta L*\right)}^{2}+{\left(\Delta a*\right)}^{2}+{\left(\Delta b*\right)}^{2}\right)}$ equation [1,4].

### 3.6. Characterizations

Elemental analyses (CHN) of the samples were performed in a Perkin-Elmer PE-2400 microelemental device. The samples were characterized by X-ray diffractometry (XRD), using a Shimadzu diffractometer (model Labx-XDR 6000, Japan) operating at 40 kV and 30 mA, with Cu Kα radiation (λ = 1.5406 Å) and 3 to 75° 2θ range with a speed of first min^−1^. The Fourier transform infrared spectroscopy (FTIR) analyses of the samples were performed in a Bruker Vertex 70 model equipment (Germany) by preparing KBr pellets at 1% (m/m). FTIR spectra were obtained with a resolution of 4 cm^−1^, 120 scans between 4000 and 400 cm^−1^. The morphology was carried out by scanning electron microscopy (SEM), using a field emission electron source (SEM-EC) device from FEI, model Quanta FEG-250 (Eindhoven, Netherlands). The samples were mounted on stubs using double-sided carbon tape and covered in gold. The SEM analysis conditions were from 8 to 20 kV, and the working distance was 10 mm with point 3. The thermal stability of the samples was studied using the thermogravimetry technique (TG). TG curves were obtained using the TGA-50 Shimadzu (Japan) device at 10 °C min^−1^ heating rate in an argon atmosphere in alumina pun at 25 to 800 °C, and a mass of approximately 8 mg. Dye solutions were scanned using a UV-Vis spectrophotometer Cary 60, Agilent Technologies (USA), between 250 and 800 nm.

## 4. Conclusions

CTAB-intercalated bentonite was used as a matrix to improve the photostability of ACN dye. The incorporation of CTAB surfactants and ACN molecules into the Bent clay sample was confirmed by the increases in the d_001_ spaces of 0.11 nm, 0.09 nm, and 0.15 nm for the pigment obtained at pH 4, 7, and 10, respectively. The new absorption bands in FTIR were attributed to the organic molecules contributions, and the morphology differences observed by the SEM images and the events of mass loss observed between 150 and 550 °C in the TG/DTG analysis were associated to the respective organic molecules incorporated. In addition, the photostability of the anthocyanin incorporated into the organoclay was confirmed by the L*a*b* colors space parameters, in which the pigment obtained at pH 10 was about 3-fold more stable than the pH 4-synthesized pigment and about 4.5-fold more stable than the other previously published.

The colors of the hybrid pigments are a function of their intermolecular interaction with ACN molecules, which is influenced by the pH used during the synthesis. Previous works demonstrated that the hybrid formation at slightly acid conditions allow the CTA^+^ species, previously incorporated into clay, to promote the conversion of the dye molecule from flavylium cation to the quinoidal base form, enabling dye incorporation and stabilizing the weakly photostable structural form of ACN. As the quinoidal base form of ACN is more pronounced in a neutral to slightly basic medium, the increase in pH used to synthesize the pigment in this work promoted stability in colors since the more vital intermolecular interaction was easily obtained under these conditions.

A wide range of possibilities is opening to develop new hybrid pigments that explore the weakly applicable colors from the quinoidal base form of anthocyanin for different application fields, exploring other clay minerals or another surfactant at a different pH of syntheses because it is a demonstrated way to stabilize this easily breakable molecule.

## Figures and Tables

**Figure 1 ijms-24-02417-f001:**
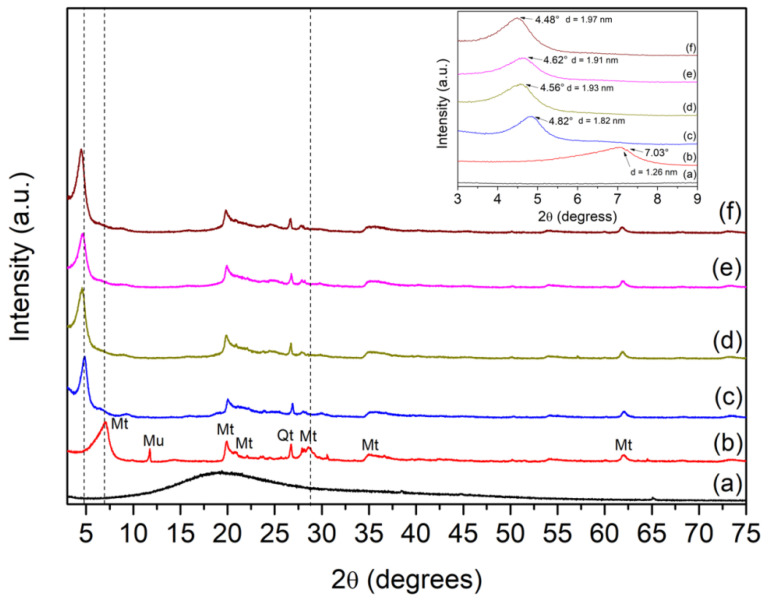
XRD patterns for ACN (**a**), Bent (**b**), CTAB/Bent (**c**), CTAB/ACN/Bent-4 (**d**), CTAB/ACN/Bent-7 (**e**), and CTAB/ACN/Bent-10 (**f**).

**Figure 2 ijms-24-02417-f002:**
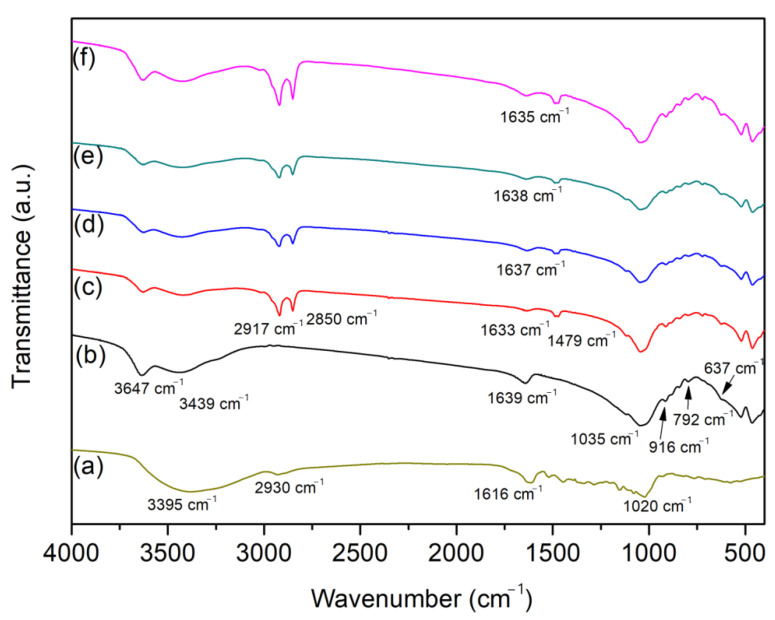
FTIR spectra of ACN (**a**), Bent (**b**), CTAB-Bent (**c**), CTAB/ACS/Bent-4 (**d**), CTAB/ACS/Bent-7 (**e**), and CTAB/ACS/Bent-10 (**f**).

**Figure 3 ijms-24-02417-f003:**
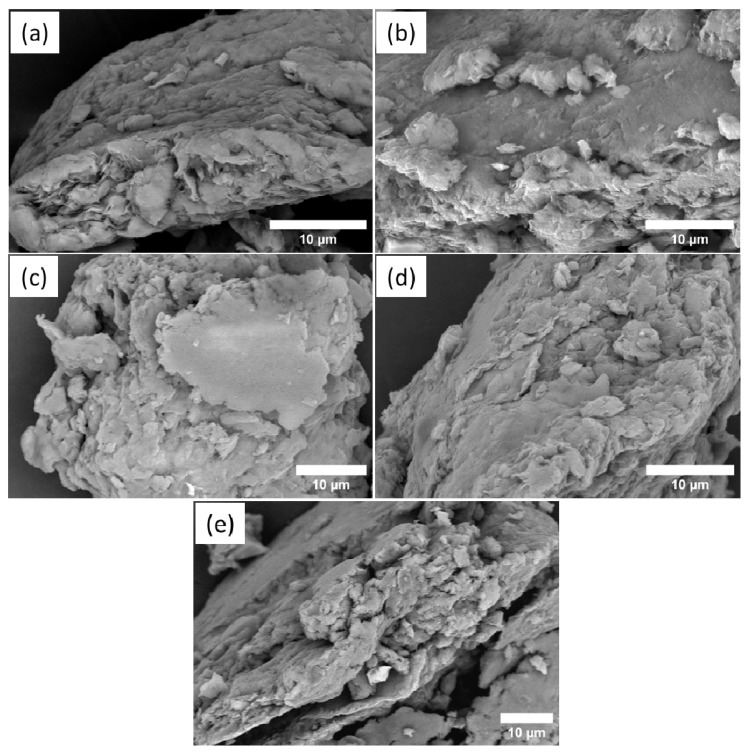
SEM of Bent (**a**), CTAB/Bent (**b**), CTAB/ACN/Bent-4 (**c**), CTAB/ACN/Bent-7 (**d**), and CTAB/ACN/Bent-10 (**e**).

**Figure 4 ijms-24-02417-f004:**
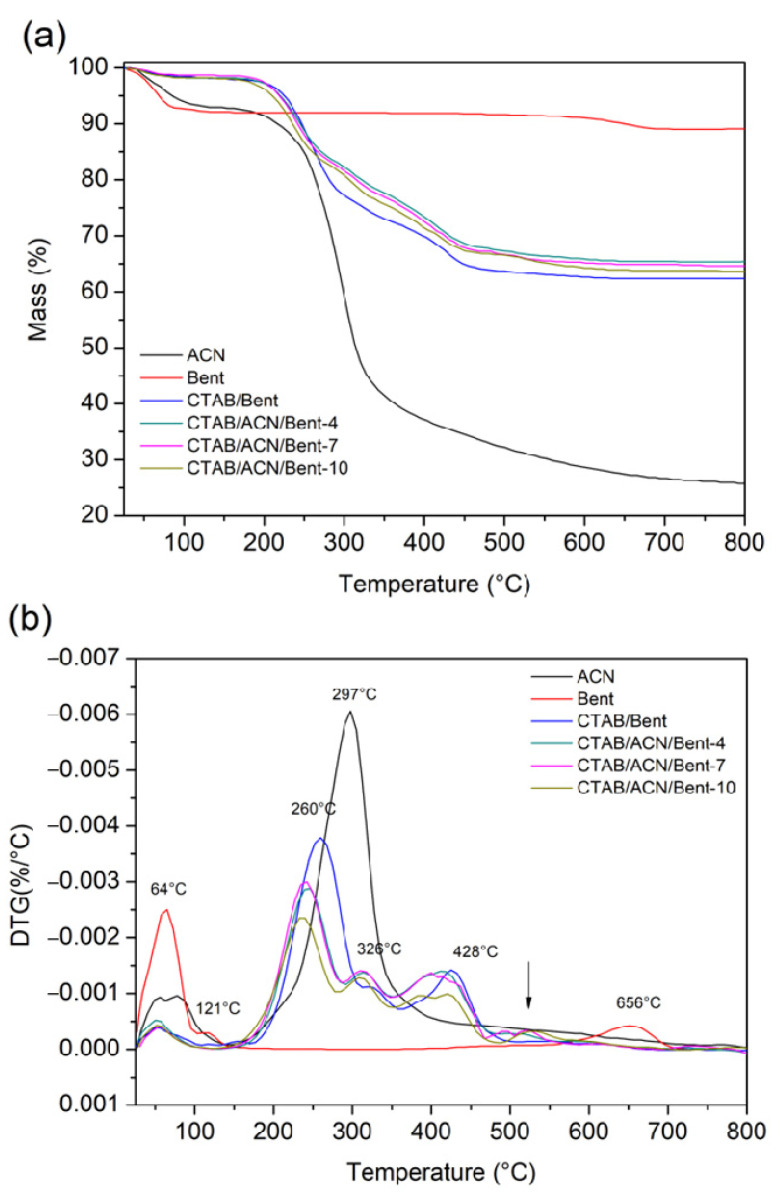
TG (**a**) and DTG (**b**) curves of ACN, Bent, CTAB/Bent, and dyed solids.

**Figure 5 ijms-24-02417-f005:**
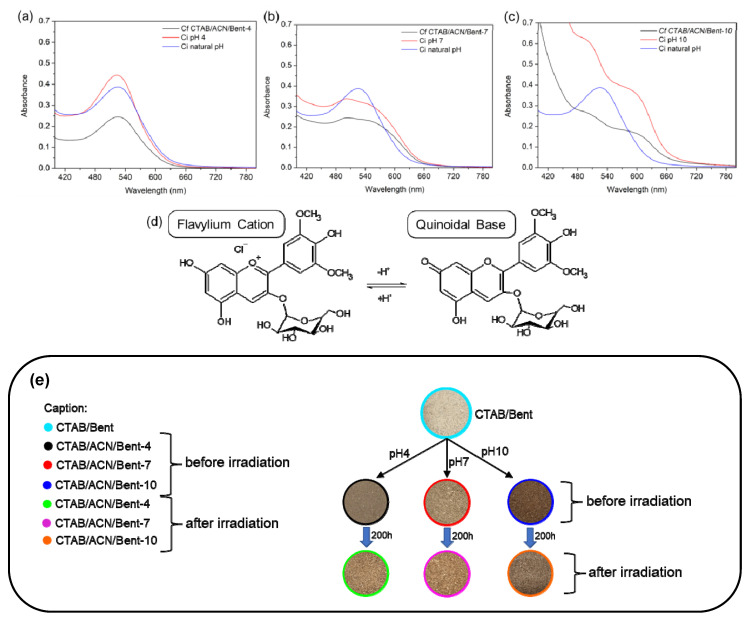
UV-Vis spectra of the ACN solutions at natural pH and pH 4 (**a**), 7 (**b**), and 10 (**c**) before and after incorporating dye in the CTAB/Bent hybrid. Ci and Cf are the initial and final dye concentrations, respectively, (**d**) deprotonation of the flavylium cations into the quinoidal base form of ACN (**e**) Digital photographs of the hybrid pigments before and after irradiation.

**Figure 6 ijms-24-02417-f006:**
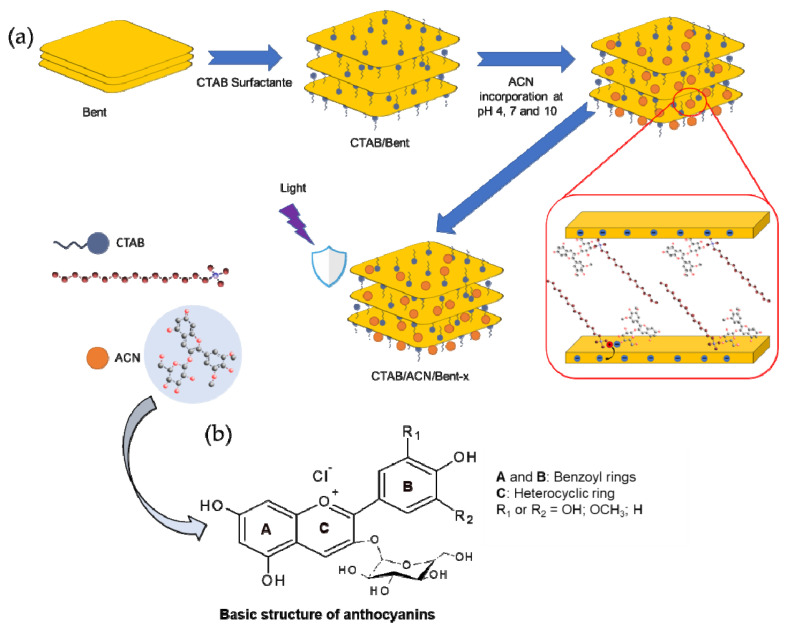
(**a**) Scheme relating the synthesis of the CTAB/ACN/Bentx hybrids and (**b**) basic structure of anthocyanins (ACN).

**Table 1 ijms-24-02417-t001:** Main thermal degradation events of Bent and its derivatives.

Sample	Degradation Event	Weight Loss [%]	Temperature Range [°C]	Maximum Degradation Temperature [°C]
**ACN**	**First event**: Referring to the removal of water from the surface	6.92%	26–132 °C	78 °C
	**Second event**: Referring to the thermal decomposition of ACN	67.20%	132–800 °C	297 °C
**Bent**	**First event**: Referring to the removal of water from the surface	7.32%	30–87 °C	64 °C
	**Second event**: Referring to the removal of water from the crystal lattice	1.00%	88–143 °C	121 °C
	**Third event**: Referring to the dihydroxylation of bentonite	1.90%	591–710 °C	656 °C
**CTAB/Bent**	**First event**: The loss of adsorbed water.	1.65%	38–95 °C	53 °C
	**Second event**: Decomposition of the CTAB physically adsorbed on the surface of the material	21.31%	200–310 °C	260 °C
	**Third event**: Decomposition of the CTAB molecules incorporated between the layers	2.72%	330–370 °C	326 °C
	**Fourth event**: Decomposition of the intercalated CTAB cations	5.72%	400–465 °C	428 °C
	**Fifth event**: Referring to the dihydroxylation of bentonite			
**CTAB/ACN/Bent-4**	**First event**: The loss of adsorbed water	1.37%	35–65 °C	50 °C
	**Second event**: Decomposition of the CTAB physically adsorbed on the surface of the material and ACN	12.91%	185–276 °C	243 °C
	**Third event**: Decomposition of the CTAB molecules incorporated between the layers and ACN	4.71%	280–328 °C	315 °C
	**Fourth event**: Decomposition of the intercalated CTAB cations and ACN	9.61%	335–336 °C	418 °C
	**Fifth event**: Referring to thedihydroxylation of bentonite	2.31%	470–645 °C	505 °C
**CTAB/ACN/Bent-7**	**First event**: The loss of adsorbed water	1.42%	35–66 °C	54 °C
	**Second event**: Decomposition of the CTAB physically adsorbed on the surface of the material and ACN	13.25%	186–269 °C	239 °C
	**Third event**: Decomposition of the CTAB molecules incorporated between the layers and ACN	5.12%	282–330%	310 °C
	**Fourth event**: Decomposition of the intercalated CTAB cations and ACN	9.94%	334–445 °C	415 °C
	**Fifth event**: Referring to the dihydroxylation of bentonite	2.53%	468–663 °C	520 °C
**CTAB/CAN/Bent-10**	**First event**: The loss of adsorbed water	1.00%	34–70 °C	52 °C
	**Second event**: Decomposition of the CTAB physically adsorbed on the surface of the material and ACN	13.29%	180–265 °C	236 °C
	**Third event**: Decomposition of the CTAB molecules incorporated between the layers and ACN	6.02%	276–326 °C	308 °C
	**Fourth event**: Decomposition of the intercalated CTAB cations and ACN	10.14%	327–452 °C	405 °C
	**Fifth event**: Referring to the dihydroxylation of bentonite	3.34%	462–662 °C	530 °C

**Table 2 ijms-24-02417-t002:** Total color differences (ΔE*) for each pigment compared with CTAB/Bent precursor and after 200 h of irradiation.

Before Irradiation					After Irradiation				
Sample	L*	a*	b*	ΔE* Compared with CTAB/Bent	Sample	L*	a*	b*	ΔE* Compared with Non-irradiated Sample
CTAB/Bent	67.7 ± 0.0	0.7 ± 0.0	11.9 ± 0.1	-	-	-	-	-	-
CTAB/ACN/Bent-4	54.3 ± 0.1	1.7 ± 0.3	8.4 ± 0.5	13.6 ± 0.2	CTAB/ACN/Bent-4	40.5 ± 0.3	4.2 ± 0.2	9.7 ± 0.3	14.1 ± 0.4
CTAB/ACN/Bent-7	47.4 ± 0.3	2.2 ± 0.4	10.0 ± 0.2	20.6 ± 0.4	CTAB/ACN/Bent-7	42.8 ± 0.2	4.5 ± 0.0	10.4 ± 0.2	5.4 ± 0.3
CTAB/ACN/Bent-10	42.4 ± 0.3	5.7 ± 0.3	12.8 ± 0.3	25.6 ± 0.2	CTAB/ACN/Bent-10	47.5 ± 0.3	4.7 ± 0.1	12.0 ± 0.4	4.2 ± 0.4

**Table 3 ijms-24-02417-t003:** Cation exchange capacity (CEC) of Bent by the ammonium acetate method.

Phases	Description
1	Initially, 1 g of clay is dispersed in 100 mL of an ammonium acetate solution with a concentration of 1.0 mol L^−1^ (pH = 7), and the system is kept under stirring for 72 h.
2	After 72 h, the solid is separated by centrifugation at 5000 rpm for 10 min (NI-1812 NOVA Instruments benchtop centrifuge, Brazil), 100 mL of ammonium acetate solution (1.0 mol L^−1^) is added again, and the system remains under agitation for another 72 h.
3	The procedure of step 2 is repeated again, and finally, the solid is washed with distilled water and ethanol, filtered off, and dried at room temperature (25 °C).
4	The CEC is calculated from the results of the CHN elemental analysis (Perkin-Elmer model PE 2400, USA) of the solid obtained.

Note: the procedure is performed in triplicate.

**Table 4 ijms-24-02417-t004:** Summary of acronyms used in the synthesized samples.

Samples	Description
Bent	Bentonite clay mineral
ACN	Anthocyanin
CTAB	Surfactant hexadecyltrimethylammonium bromide
CTAB/Bent	Bentonite with CTAB
CTAB/ACN/Bent-4	Adsorption of CTAB/Bent with ACN solution at pH 4
CTAB/ACN/Bent-7	Adsorption of CTAB/Bent with ACN solution at pH 7
CTAB/ACN/Bent-10	Adsorption of CTAB/Bent with ACN solution at pH 10

## Data Availability

Not applicable.

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
