# Peer review of "Organic–Inorganic Hybrid Pigments Based on Bentonite: Strategies to Stabilize the Quinoidal Base Form of Anthocyanin"

_ijms, 2023, doi:10.3390/ijms24032417_

Round 1
Author Response
Dear Reviewer 1,
Thank you very much for your attention and for the reviewers’ comments on our manuscript ‘Organic-inorganic hybrid pigments based on bentonite: Strategies to stabilize the quinoidal base form of anthocyanin’ (Manuscript ijms-2050280). We agree with the comments and suggestions, which have been of great assistance in improving the quality of our paper and guiding our research. We have revised the paper after carefully studying the reviewer’s comments and have responded to them point by point. Revisions to the manuscript are highlighted in red color. The main revisions and detailed responses to the referees’ comments are listed below.

Reviewer 2 Report
The authors reported new functional materials of using cetyltrimethylammonium bromide in bentonite clay for incorporation and photostabilization of anthocyanin dye, and sufficiently characterized its synthesis, structure (elemental analysis, XRD and IR spectroscopy). Furthermore, Visible light-driven photostability tests were performed to confirm that the organic-inorganic matrices were efficient to stabilize the quinoidal base form of anthocyanin. After the carefully evaluation, I do consider that it is very interesting to the readers of INTERNATIONAL JOURNAL OF MOLECULAR SCIENCES, whereas, the following aspect should be promoted after minor revision.
1,Figure 5d and Figure 7b are not so clear, please redraw them.
2, Please check the chemical formula of R1=R2…..in Figure 7d,
3, page 10,line 259, pHs is right? Table 3, phases 1, pH7? Or pH=7?
4, ref8, no page number, please check the ref format carefully.
5, the SEM pictures in Figure 3 are not so homogeneous, please explain or supply high-quality pictures
Author Response
Dear Reviewer 2,
Thank you very much for your attention and for the reviewers’ comments on our manuscript ‘Organic-inorganic hybrid pigments based on bentonite: Strategies to stabilize the quinoidal base form of anthocyanin’ (Manuscript ijms-2050280). We agree with the comments and suggestions, which have been of great assistance in improving the quality of our paper and guiding our research. We have revised the paper after carefully studying the reviewer’s comments and have responded to them point by point. Revisions to the manuscript are highlighted in red color. The main revisions and detailed responses to the referees’ comments are listed below.

Reviewer 3 Report
The author's paper entitled “Organic-inorganic hybrid pigments based on bentonite: Strategies to stabilize the quinoidal base form of anthocyanin” is timely.
I would recommend the suggestions described below:
Abstract should be quantitative as possible for rapid comparison with others studies, referring for instance to the percentage of the increase (how much?). Avoid imprecise terms such more stable or more easly…. but how much? 2-fold? 50%? Regarding the control?. Verify ref 8, that is incomplete. Introduction: at the end of the intro, it is also not clear what is the main message and relevant points and highlights of the paper that should be emphasize at this stage. What is new and timely in the present study? Avoid imprecise terms along the text such as significantly and/or significant (reduced; changes): but how much? Was the data (for instance from Table I and Table II) validated by statistical analysis? Please, clarify. The conclusion should followed the order of presentation of the paper and resume partial and then global conclusions and also with perspectives for future research.
Author Response
Dear Reviewer 3,
Thank you very much for your attention and for the reviewers’ comments on our manuscript ‘Organic-inorganic hybrid pigments based on bentonite: Strategies to stabilize the quinoidal base form of anthocyanin’ (Manuscript ijms-2050280). We agree with the comments and suggestions, which have been of great assistance in improving the quality of our paper and guiding our research. We have revised the paper after carefully studying the reviewer’s comments and have responded to them point by point. Revisions to the manuscript are highlighted in red color. The main revisions and detailed responses to the referees’ comments are listed below.
